# SCALABLE TRANSFER LEARNING
# WITH EXPERT MODELS

**Joan Puigcerver**[*]
Google Research

**Carlos Riquelme**[*]
Google Research

**Basil Mustafa**
Google Research

**Cedric Renggli**[†]
ETH Zurich

**André Susano Pinto**
Google Research

**Sylvain Gelly**
Google Research

**Daniel Keysers**
Google Research

**Neil Houlsby**
Google Research

## ABSTRACT

Transfer of pre-trained representations can improve sample efficiency and reduce computational requirements for new tasks. However, representations used for transfer are usually generic, and are not tailored to a particular distribution of downstream tasks. We explore the use of expert representations for transfer with a simple, yet effective, strategy. We train a diverse set of experts by exploiting existing label structures, and use cheap-to-compute performance proxies to select the relevant expert for each target task. This strategy scales the process of transferring to new tasks, since it does not revisit the pre-training data during transfer. Accordingly, it requires little extra compute per target task, and results in a speed-up of 2–3 orders of magnitude compared to competing approaches. Further, we provide an adapter-based architecture able to compress many experts into a single model. We evaluate our approach on two different data sources and demonstrate that it outperforms baselines on over 20 diverse vision tasks in both cases.

## 1 INTRODUCTION

Deep learning has been successful on many computer vision tasks. Unfortunately, this success often requires a large amount of per-task data and compute. To scale deep learning to new vision tasks, practitioners often turn to transfer learning. Transfer learning involves re-using models trained on a large *source* task, and tuning them on the *target* task. This can improve both convergence rates (Ben-David et al., 2007; 2010; Blitzer et al., 2008; Du et al., 2017; Kuzborskij & Orabona, 2013; Mansour et al., 2009) and empirical performance (Dai et al., 2007; Donahue et al., 2014; Oquab et al., 2014; Tan et al., 2018). Transfer learning reduces *per-task* data or compute requirements, given a large one-off pre-training cost. In practice, this one-off down payment may not be made by the practitioner, since pre-trained networks are made available through platforms like PyTorch and TensorFlow Hub[1]. For instance, ImageNet pre-training is popular since it is freely available and works well for many tasks (Donahue et al., 2014; Oquab et al., 2014; Sharif Razavian et al., 2014).

In contrast to generic homogeneous models (e.g. most pre-trained ImageNet networks), Mixture of Experts (MoE) include multiple heterogeneous sub-models ("experts") that specialize to sub-problems of the full task. MoEs have been studied for decades (Eigen et al., 2013; Jacobs & Jordan, 1993), and have also been successful in deep learning (Shazeer et al., 2017). Yet, the application of experts for deep transfer learning has been less explored. We study visual transfer with experts, and present a simple, scalable, yet effective strategy.

Transfer of specialist models has been studied before. However, they either require expensive re-training on the source dataset for every target task (Ngiam et al., 2018; Yan et al., 2020), or operate at a small scale where all experts can be applied simultaneously (Dvornik et al., 2020). Further, most of them are tested only on a limited suite of natural single-object classification tasks. We lift these

---

[*]Equal contribution. Order decided by a coin toss.
[†]Work done while interning at Google Research.
[1]`https://pytorch.org/hub/` and `https://tfhub.dev/`, respectively.

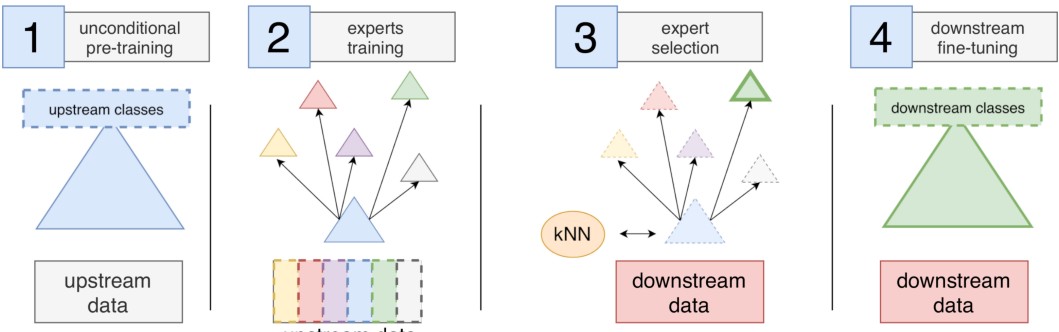

Figure 1: Transfer Learning with Per-Task Routing of Experts. **Step 1.** A single baseline model **B** is trained on the entire upstream dataset. **Step 2.** The upstream data is divided in semantic subsets (possibly overlapping). One expert is trained on each subset using the weights from **B** as initialization. **Step 3.** Given a new downstream task $\mathbf{D}_T = (X_T, Y_T)$, we compute the image representations $M_e(X_T)$ from each expert $e$. We use kNN to compute the accuracy on the supervised problem $\mathbf{D}_{T,e} = (M_e(X_T), Y_T)$, and select the expert $e^*$ with highest accuracy. **Step 4.** We add a new head to $e^*$ and fine-tune its whole network with the downstream data, leading to the final model.

constraints, and present a practical approach that scales to hundreds of large experts, while requiring relatively little compute *per target task*.

Our strategy consists of four stages (fig. 1). (1) *Unconditional pre-training*. A single baseline model is trained on the entire upstream data. (2) *Experts training*. Multiple experts are pre-trained by exploiting the label hierarchy present in many large-scale image datasets, such as ImageNet and JFT. In addition to entire expert networks, we explore residual adapters that allow all of the expertise to be packed into a single model that can be loaded into memory. These two stages may be expensive, but are done only once. (3) *Expert selection*. Applying all experts to each task does not scale well; some sort of sparsification is required. We focus on inexpensive model selection that can be applied to hundreds or thousands of experts. (4) *Downstream fine-tuning*. We take the output of the model selection phase and tune it on the target task. Importantly, this phase does not require revisiting the source dataset, which may be unavailable or expensive to train on.

We show that this approach yields remarkably strong performance on many diverse tasks. We evaluate not only on classic vision tasks, but also on the diverse VTAB benchmark of 19 tasks (Zhai et al., 2019). Our contributions can be summarized as follows.

- We propose a transfer learning algorithm with a large number of experts based on per-task routing via nearest neighbors selection. Once we have amortized the pre-training cost, this algorithm requires little compute *per target task*, achieving an speed-up of $500\times$–$1000\times$ compared to competing strategies. Also, it can be easily replicated with any large upstream multilabel dataset.

- We achieve a mean accuracy improvement of 3.6% over the state-of-the-art performance on 19 VTAB datasets using ResNet50 networks. Our algorithm offers improvements on every group of tasks: natural, specialized, and structured. Figure 2 summarizes these results.

- We explore using sub-networks as experts via residual adapters, allowing all experts to be packed into a single model. Surprisingly these perform almost as well as their full-network counterparts.

## 2 RELATED WORK

**Transfer Learning.** Tasks with little training data can benefit from other larger datasets, often from a similar domain. Transfer learning concerns the link between the source and target dataset (Pan & Yang, 2009; Weiss et al., 2016; Tan et al., 2018; Wang, 2018). One family of methods creates a single training dataset, where source instances are re-weighted according to their relevance (Dai et al., 2007; Pardoe & Stone, 2010; Wan et al., 2011; Xu et al., 2017). A popular method consists of fine-tuning a model that was pre-trained on the source data (Donahue et al., 2014; Oquab et al., 2014; Sharif Razavian et al., 2014). Some transfer learning algorithms condition the initial source model on the target dataset itself (Ngiam et al., 2018; Xie et al., 2019; Yalniz et al., 2019), while

Figure 2: Summary of results on the VTAB-1k benchmark, combining experts with different architectures trained on two different data sources (JFT, ImageNet21k). In each of the 19 datasets, we use the median accuracy over 30 runs. The average of the accuracies in each group is shown, as well as (percentile) bootstrap confidence intervals at the 95% level.

others (like ours) are agnostic about the downstream task when the initial model is trained on the source data (Kolesnikov et al., 2019). We offer an in-depth comparison with (Ngiam et al., 2018) in section 6.6. In the context of few-shot learning, where out-of-the-box fine-tuning may not work, generic representations are sometimes frozen, and simple feature selection (Dvornik et al., 2020) or model training (Chen et al., 2019) techniques are applied on top. Instead of relying on fixed universal representations, (Rebuffi et al., 2017; 2018) use small additional modules, or adapters, that incorporate knowledge from several visual domains. Our work also explores this idea.

**Multi-task Learning.** MTL tries to leverage the common aspects of several learning tasks (Caruana, 1997). A prominent approach uses explicit parameter sharing; for instance, by means of common low-level layers leading to different heads. Among others, this has been successfully applied to vision (Zhang et al., 2014), language (Liu et al., 2015), and reinforcement learning (Fedus et al., 2019) tasks. In addition, a variety of ways to combine task-specific representations have arisen, such as cross-stitch networks (Misra et al., 2016), or lateral connections (Rusu et al., 2016). A different family of methods impose joint constraints on the –possibly different– models corresponding to each task. We can combine the learning problems via regularization and shared sparsity patterns (Argyriou et al., 2007; Lounici et al., 2009), or by imposing some prior knowledge regarding the task structure (Evgeniou et al., 2005; Jacob et al., 2009; Kim et al., 2012).

## 3 THE TRANSFER LEARNING FRAMEWORK

In this section, we describe our transfer learning setup of interest. The high-level goal is to train strong models for arbitrary downstream tasks, possibly under severe data and compute limitations. To do so efficiently, one can offload computation to a previous upstream phase which is executed *a priori*, without knowing the downstream tasks in advance. Accordingly, the upstream model should not depend on any specific target data. We are mostly interested in the *low data* regime where downstream tasks contain few datapoints. These restrictions have a practical motivation: we would like to build and deploy universal representations that are easily transferred to a wide range of downstream settings. Any transfer algorithm must implement the following three stages.

**Upstream Training.** Given the upstream data $\mathbf{D}_U$, the algorithm first outputs a source model $\mathbf{M}$. The goal is to provide useful initial representations for various new tasks. This stage could actually produce a family of models $\{\mathbf{M}_e\}$ rather than a single one. These models might not be disjoint, and could share parameters. The upstream learning problems are auxiliary; accordingly, $\mathbf{D}_U$ could include a diverse set of classification, regression, or even synthetic learning instances.

**Model Selection.** When a new downstream task is given, a selection algorithm is applied to choose the upstream model(s) to transfer, possibly depending on the downstream data. This phase should be computationally cheap; thus, the upstream data is no longer available. Sometimes, there is no choice to make (say, with a single ImageNet representation). Alternatively, in models with a complex structure, one may choose which parts, routes, or modules to keep in a data-dependent fashion.

**Downstream Training.** The final stage fine-tunes the selected model using the downstream data, either fully or partially. For neural nets, a new head is added as the output classes are task-specific.

Our overall algorithm is depicted in fig. 1. We give details about each step in the following sections.

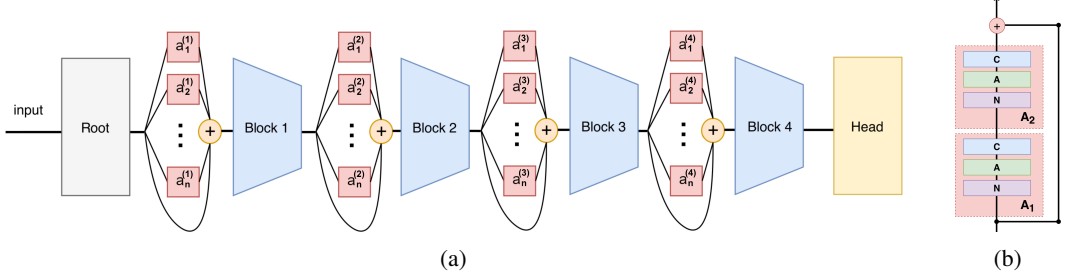

(a)  (b)

Figure 3: (a) ResNet with expert adapters before all blocks. A layer of experts is placed before every block. (b) Each individual adapter including the overall skip connection. N, A, C stand for (Group) Normalization, (ReLU) Activation, and Convolution layers, respectively.

# 4 UPSTREAM TRAINING

## 4.1 EXPERT ARCHITECTURES

Our experts should provide feature extractions that are a good starting point to learn future tasks related to the expert's upstream training data. We explore two different model architectures to train such experts. As an obvious choice, we first look at ResNets (He et al., 2016b). These are powerful models; however, storing and deploying many of them can be challenging. As an alternative, we also develop more compact adapter modules that can all be assembled in a single architecture. Also, their individual size can be easily customized to meet memory and computational constraints, which makes them an ideal candidate for combining multiple experts in a single model, when needed. We informally refer to these as *full* and *adapter* modules (or experts), respectively.

**Full ResNet Modules.** As a base architecture for full experts we use ResNets. In particular, all of our experiments focus on the ResNet50-v2 architecture (R50) (He et al., 2016a), which sequentially stacks a root block and 4 blocks with (3, 4, 6, 3) residual units. The initial step in every experiment consists of training a baseline model **B** on the whole upstream data (see stage 1 in fig. 1). This baseline is subsequently fine-tuned by both full and adapter experts, but in different ways. A full expert trained on a slice of data is simply the baseline **B** fine-tuned on that data. The head will later be discarded for transfer. This approach requires as many R50s as there are experts.

**Adapter Modules.** Residual adapters were proposed to adapt a neural network to a particular downstream task without needing to fine-tune the entire network (Rebuffi et al., 2017). Instead, we use them to adapt the baseline architecture to slices of the *upstream data*. Originally, they were $1 \times 1$ convolutions placed after each $3 \times 3$ convolution, with a residual connection. Instead we place them before each of the R50's blocks. Finally, our adapters have a bottleneck and are non-linear, as in (Houlsby et al., 2019). We insert several in parallel into the backbone **B**. When creating an expert, only the adapters are tuned and the backbone weights are frozen.

Figure 3a depicts the ResNet architecture with multiple expert adapters $(a_1^{(i)}, \ldots, a_n^{(i)})$. Let $F_i$ be the function implemented by the $i$-th block of the backbone network. We *adapt its input* by computing the output as $x_i := F_i(x_{i-1} + a_e^{(i)}(x_{i-1}))$, where $e = R(x)$ is the identity of the selected expert, given by some routing function $R$, and $x$ is the original input. During upstream training, the function $R$ may also use the labels in addition to the image, as we discuss in section 4.3.

Figure 3b shows the adapter's bottleneck architecture. An adapter sequentially applies components $A_1$ and $A_2$. Each component performs a group normalization (N) (Wu & He, 2018), a ReLU activation (A) (Glorot et al., 2011), and a convolution (C) (LeCun et al., 1989), in that order. Due to the skip connection, the output dimension of $A_2 \circ A_1$ must match that of its input, $c$. However, we can change the output channels $k$ of $A_1$, in order to limit the amount of parameters. Thus, we set $k = \frac{c}{2}$ so that the number of parameters equals that of a linear adapter. Each adapter only increases the parameter count of the R50 backbone by 6%. We briefly explored placing these adapters in other locations, or using other variations (Rebuffi et al., 2018), but we did not observe any significant improvement.

It is important to emphasize that the names *Full* and *Adapters* refer to the parameters that are specialized to a given subset of the upstream data *before any downstream fine-tuning*. In both cases, we fine-tune the entire network, after the model selection phase, to the downstream task.

## 4.2 UPSTREAM DATA AND EXPERT DEFINITION

We train our upstream models on large vision datasets with thousands of classes. Moreover, the datasets include an expressive hierarchy, linking classes and ancestor concepts via "is-a" relationships. Our experts' domains are nodes in this hierarchy, which are selected automatically based on the number of images. Due to the multi-label nature of the datasets, several experts could simultaneously apply to an image. For example, for an image of a lion, all of *organism, animal, carnivore, felidae*, and *lion* could be relevant expert domains. In particular, we use two different upstream image datasets, and independently train a set of experts on each. We further describe them in section 6.1.

## 4.3 EXPERT TRAINING

Recall we denote by $\mathbf{B}$ the baseline R50 model trained on the whole upstream dataset $\mathbf{D}_U$. As shown in fig. 1, the second step of upstream training consists of training each expert individually on different subsets of the upstream dataset. Let $\mathbf{D}_e := (X_e, Y_e) \subseteq \mathbf{D}_U$ be the data corresponding to expert $e$. The subsets corresponding to different experts may overlap (e.g. for the *animal* and *dog* experts).

As mentioned before, the *full* experts completely fine-tune $\mathbf{B}$ on $\mathbf{D}_e$. For the *adapter* experts the weights corresponding to the adapter $e$ (modules in red in fig. 3) are trained on $\mathbf{D}_e$, but the shared blocks and head parameters are *frozen*. Note that, due to the sharing scheme, we can train all experts independently in parallel. We train all experts for the same number of steps, regardless of the size of $\mathbf{D}_e$. Instead of learning a routing function, we exploit the structure of the upstream labels and use a hard-coded routing. We found this makes learning easier, and leads to powerful specialized models.

## 5 EXPERT SELECTION

Given a new downstream dataset $\mathbf{D}_T = (X_T, Y_T)$, we must choose an expert to use. We consider three approaches: *domain prediction*, *label matching*, and *performance proxy*.

**Domain Prediction.** This strategy selects the expert solely based on the images $X_T$. It effectively selects the expert whose domain best matches the target images. We implement this by training an auxiliary network (the "Expert Prediction Network" or EPN) to classify the expert from the image (i.e. learn the hard-coded routing mentioned previously). The EPN is trained upstream using the pre-training data and expert assignments. During transfer, an expert is selected using the highest geometric mean EPN predictive probability across the dataset. Details are in the Appendix A.

**Label Matching.** Matching the task with an expert can be done in the label space as opposed to the input space, by computing the affinity of each expert to a new downstream task in the label space of the upstream dataset. We first use a generic network trained on all upstream labels to predict upstream labels on the downstream images. We compute the KL-divergence between the distribution of labels on the downstream task images, and the prior distribution of labels for each expert. This per-expert prior is computed as the empirical distribution of labels on the images used to train that expert. We select the expert with the smallest KL-divergence. Details are in the Appendix B.

**Performance Proxy.** The aforementioned two strategies are simple, but do not use the training labels $Y_T$ available for downstream tasks, which may contain key information. It would be too expensive to fine-tune every expert to every new task and select the best with hindsight, so we propose a proxy for the final performance. For this, we use a $k$-nearest neighbors classifier (Altman, 1992) with the image embeddings produced by each expert. In the case of full experts, we simply apply the corresponding full network to compute these embeddings. For adapter-based experts, we apply the specific expert and ignore the remaining ones. Concretely, let $M_e(x)$ be the embedding corresponding to expert $e$ on input $x$, and let $\mathbf{D}_T = \{(x_i, y_i)_{i=1}^{N_T}\}$ be our downstream task. In order to score each expert, we apply a kNN classifier on the embedded dataset $\mathbf{D}_{T,e} = \{(M_e(x_i), y_i)_{i=1}^{N_T}\}$, with $k = 1$ and Euclidean distance. The accuracy $\mathrm{acc}(\mathbf{D}_{T,e})$ is computed via leave-one-out cross-validation. Finally, we select the expert with highest accuracy: $e^* = \arg\max_e \mathrm{acc}(\mathbf{D}_{T,e})$. There are other alternative proxies that

are cheaper than full fine-tuning, for example fitting a logistic regression, SVM, or decision trees to the features. Although these proxies may better match the fine-tuning accuracy, we use kNN since it is computationally cheap — it only requires a forward pass through the data, and leave-one-out cross-validation requires no additional inference per-fold — and it performs well (section 6).

## 5.1 DOWNSTREAM TRANSFER

The expert selection algorithm could choose several experts to be combined to solve any target task. However, we limit the scope of our work to transferring a single expert per task, since this approach is simple and turns out to be effective. Thus, the downstream transfer procedure is straightforward: it simply involves fine-tuning the selected expert model. We fine-tune the entire expert network to the downstream dataset, including the adapters when applicable. While it was valuable to restrict the scope of upstream training to focus on specializing the expert adapter parameters, we found fine-tuning the whole network downstream to be greatly beneficial. This differs from the original residual adapters work (Rebuffi et al., 2017), where only the adapters were fine-tuned.

## 6 EXPERIMENTAL RESULTS

### 6.1 UPSTREAM TRAINING

**ImageNet21k** (Deng et al., 2009) is a public dataset containing 13 million images, and 14 million labels of 21 843 classes, which are WordNet synsets (Fellbaum, 2012). In addition to the 21k classes, we consider the 1 741 synsets that are their ancestors. We use the 50 synsets of ImageNet21k with the largest number of images to train the expert models. These include e.g. *animal, artifact, organism, food, structure, person, vehicle, plan*, or *instrument*. We released 48 of these ImageNet21k models[2].

**JFT** (Sun et al., 2017) is an even larger dataset containing 300 million images and 18 291 classes. Each image can belong to multiple classes, and as for ImageNet21k, the classes are organized in a hierarchy. We select as expert domains the classes with a sufficiently large number of examples: the 240 classes with more than 850 000 images. Some of the automatically chosen experts are *animal, arts, bird, food, material, person, phenomenon, plant*, or *product*.

We pre-train generic models on a Cloud TPUv3-512, as done in (Kolesnikov et al., 2019). Then fine-tune them briefly on each slice to create the expert models. Check additional details in appendix D.

### 6.2 DOWNSTREAM TASKS

We evaluate on two suites of tasks, each consisting of several datasets. The first is the Visual Task Adaptation Benchmark (VTAB) (Zhai et al., 2019), which consists of 19 datasets. We evaluate on VTAB-1k, where each task contains only 1k training examples. The tasks are diverse, and divided into three groups: natural images (single object classification), structured tasks (count, estimate distance, etc.), and specialized ones (medical, satellite images). Appendix E.1 contains further details. The second suite is a collection of popular natural datasets commonly used in transfer learning literature and, particularly in (Ngiam et al., 2018), to which we compare our work.

### 6.3 TRANSFER EVALUATION PROTOCOL

When transferring to new tasks we need to perform expert selection and choose other hyperparameters (e.g. learning rate for fine-tuning). For each downstream task, we use the following three step protocol.

**Expert Transfer**. We select the expert to transfer using one of the methods presented in section 5. In both sets of tasks, we use 1k training examples per dataset. Details are provided in appendix C.1.

**Hyperparameter Selection**. In VTAB-1k we use the recommended hyperparameter sweep and 800-training/200-validation split. We independently repeat the hyperparameter selection procedure 10 times for confidence intervals. For the other datasets we perform a single random search over 36 hyperparameter sets and select the best set based on the validation performance. This is a similar computational budget to that of (Ngiam et al., 2018). See appendices E.2 and F.1 for sweep details.

---

[2]https://tfhub.dev/google/collections/experts/bit/1

Table 1: VTAB-1k results of different selection algorithms, using full experts trained on JFT. The average accuracy across each group of tasks and across all VTAB is reported. In each dataset, the median accuracy over 30 runs is used. Bootstrapped confidence intervals at 95% level are included.

| | NATURAL | SPECIALIZED | STRUCTURED | ALL |
|---|---|---|---|---|
| Baseline (No Experts) | 77.4 [77.3–77.6] | 81.6 [81.5–82.0] | **57.2** [52.8–58.2] | 69.8 [68.0–70.2] |
| Random Expert | 60.6 [59.1–63.9] | 81.2 [80.9–81.8] | 56.8 [54.9–57.8] | 63.3 [62.3–64.6] |
| Domain Prediction | 75.9 [74.4–77.4] | 81.5 [81.3–82.2] | 57.0 [56.1–57.4] | 69.1 [68.4–69.8] |
| Label Matching | 77.6 [77.8–78.1] | 80.3 [79.1–82.5] | 56.9 [55.6–57.2] | 69.6 [68.9–70.0] |
| Performance Proxy | **79.7** [79.5–80.0] | **83.6** [83.3–83.8] | 55.3 [52.1–56.3] | **70.2** [68.9–70.6] |

Table 2: VTAB-1k results of the baseline models and different expert architectures using kNN selection, pre-trained on ImageNet21k (IN21k) and JFT. The average accuracy across each group of tasks and across all 19 tasks is shown. In each dataset, the median accuracy over 30 runs is used.

| | | NATURAL | SPECIALIZED | STRUCTURED | ALL |
|---|---|---|---|---|---|
| IN21k | Baseline | 77.7 [77.4–77.8] | 82.0 [78.4–83.9] | 56.8 [55.9–57.2] | 69.8 [68.8–70.3] |
| | Adapters | 78.1 [78.0–78.3] | 83.5 [83.1–83.6] | 57.5 [56.8–58.2] | 70.6 [70.3–70.9] |
| | Full | 78.3 [78.1–78.6] | 83.4 [83.2–83.6] | 59.4 [58.7–59.8] | 71.4 [71.1–71.6] |
| | All Experts | 78.3 [78.1–78.6] | 83.6 [83.4–83.7] | 58.8 [58.0–59.4] | 71.2 [70.8–71.5] |
| JFT | Baseline | 77.4 [77.3–77.6] | 81.6 [81.5–82.0] | 57.2 [52.8–58.2] | 69.8 [68.0–70.2] |
| | Adapters | 79.0 [78.6–79.1] | 81.3 [79.2–82.5] | 59.1 [58.3–60.1] | 71.1 [70.5–71.6] |
| | Full | 79.7 [79.5–80.0] | 83.6 [83.3–83.8] | 55.3 [52.2–56.2] | 70.2 [68.9–70.6] |
| | All Experts | 80.0 [79.2–80.4] | 83.7 [83.6–83.8] | 58.6 [58.0–59.4] | 71.8 [71.3–72.2] |
| IN21k + JFT | All Experts | **80.2** [79.8–80.3] | **84.0** [83.7–84.2] | **59.5** [58.7–60.1] | **72.3** [71.9–72.6] |

**Final Re-training**. Using the hyperparameters from the previous step, we re-train the selected expert on the entire task (training plus validation set). In VTAB-1k, we repeat this step 3 times for each of the 10 trials of hyperparameter selection and compute the test accuracy, yielding 30 outcomes per method per task. We compute the median of these 30 outcomes as the final accuracy in the dataset.

### 6.4 PERFORMANCE OF DIFFERENT EXPERT SELECTION STRATEGIES

We first establish which of the expert selection strategies presented in section 5 performs best. As a baseline we also try selecting a random, uniformly drawn, expert per task. Table 1 shows the results on VTAB-1k, using full experts trained on JFT. Table 5 show the results with adapters.

Overall, all methods perform better than random selection, particularly on the NATURAL group. This confirms that selecting good experts is essential. Overall, the performance proxy (kNN) selection performs better than the other alternatives. kNN's average accuracy is 11% (relative) and 5.5% higher than that of the domain prediction and label matching, respectively. Thus, making use of the downstream labels offers a significant advantage in expert prediction. Therefore, in all subsequent experiments we use the kNN-based selection. We did not see a strong difference for the STRUCTURED datasets. We provide an extensive analysis of the kNN accuracy distribution per expert in appendix C. Appendix G shows how training experts on *random* subsets of the upstream data does not work well.

### 6.5 RESULTS ON VTAB

Table 2 shows the average accuracy across all the 19 VTAB-1k datasets broken down by type (natural, specialized, and structured). We summarize our findings as follows:

**Improvement over Non-expert Baseline.** All the algorithms, trained on either JFT or ImageNet21k, improve their corresponding baseline. Differences are most pronounced on the NATURAL datasets. While we also see improvements in SPECIALIZED and STRUCTURED datasets, some of the confidence intervals overlap. The performance of both JFT and ImageNet21k models is fairly similar in general. This is not unexpected; it has been observed before that, with restricted model capacity, they perform very similarly (Kolesnikov et al., 2019). Appendix C.6 shows the selected experts.

Table 3: Accuracy on the datasets used by DAT (Ngiam et al., 2018), and their average accuracy. Bootstrapped confidence intervals at 95% level are shown next to the accuracy where available. DAT uses Inception-v3 (In-v3) and a larger network, AmoebaNet-B (Am-B). Models pre-trained on JFT.

| | AIRCRAFT | BIRDS | CARS | CIFAR10 | FOOD | PETS* | AVG. |
|---|---|---|---|---|---|---|---|
| Baseline | 91.4 [91.0–91.7] | 78.8 [78.0–79.4] | 95.6 [95.4–95.7] | 97.8 [97.7–97.9] | 91.3 [91.2–91.5] | 94.5 [94.4–94.6] | 91.6 [91.4–91.7] |
| Adapters | 92.5 [92.2–92.8] | 79.4 [78.7–80.1] | 95.9 [95.8–96.0] | 97.9 [97.8–98.0] | 91.6 [91.5–91.7] | 94.6 [94.4–94.8] | 92.0 [91.9–92.1] |
| Full | **94.8** [94.5–95.1] | 83.6 [83.1–83.9] | **96.1** [96.0–96.3] | 97.8 [97.7–97.9] | 93.1 [92.8–93.2] | **97.0** [96.9–97.1] | 93.7 [93.6–93.8] |
| DAT (In-v3) | 94.1 | 81.7 | 95.7 | 98.3 | 94.1 | **97.1** | 93.5 |
| DAT (Am-B) | 92.8 | **85.1** | 95.8 | **98.6** | **95.3** | 96.8 | **94.1** |

*Pets results are mean per class accuracy, not accuracy.

**Quality of Natural Representations.** The upstream datasets used to train the experts mostly contain natural images. Consequently, the spectrum of representations offered by our models seem very effective in downstream natural datasets. More concretely, all models lead to improvements over the baseline performance, with average gains ranging from 1% to over 3.3% on the 7 natural datasets.

**Full vs. Adapters.** *JFT Experts.* Full models outperform adapters convincingly in NATURAL and SPECIALIZED datasets. However, they do a poor job on STRUCTURED datasets –mainly due to the failure on one specific dataset. *ImageNet21k Experts.* In this case, the advantage of full experts comes precisely from STRUCTURED datasets. Appendix E provides results broken down by each dataset.

**Combining All Experts.** The previous results suggest adding all types of expert models to the selection pool (full or adapter, trained on JFT or ImageNet), since the optimal expert architecture and upstream data vary per task. Results are remarkable: kNN is able to select good candidates to fine-tune from a pool of almost 600 models, and the relative improvement over the Baseline accuracy across all VTAB datasets is 3.6%, showing gains on all dataset types (see last row in table 2).

## 6.6 OUR APPROACH VS. DOMAIN ADAPTIVE TRANSFER

Domain Adaptive Transfer (Ngiam et al., 2018) (DAT) also relies on specialist models pre-trained on JFT. First it trains a generalist model on the upstream data, similar to our **B**. For any new task, then re-weights the *upstream* images based on a forward pass on the downstream data, and fine-tunes a new specialist model using the re-weighted upstream data. Finally, the model is further tuned on the target task. DAT falls outside of our transfer setup presented in section 5, as the downstream data *directly* influences the upstream training. This incurs a significant cost learning *every* new target task.

Remarkably, our algorithm works in setups where access to upstream data is not available (e.g. for privacy or proprietary reasons). We also use downstream labels, which proved to carry key information about the task (see section 6.4). And most importantly, our method is more practical by amortizing the cost of expert pre-training as more downstream tasks are served. Under same models and hardware, running kNN (with 240 models) is between $500\times-1000\times$ faster than fine-tuning the baseline model with the re-weighted upstream data. Appendix F includes additional details.

Table 3 shows the mean accuracy over 30 trials per dataset, on the same datasets and under a similar hyperparameter budget as DAT. These tasks are close to VTAB's NATURAL group and yield similar results: full experts outperform adapters. Our results are not directly comparable to those of DAT since they use Inception-v3 (Szegedy et al., 2016), and AmoebaNet-B (Real et al., 2019) architectures. Inception-v3 and R50 are similar in performance and size; the former has 24M parameters, attaining 78.8% top-1 on ILSVRC2012 (from-scratch), whereas the latter has 26M parameters and attains 76.0%. The AmoebaNet-B (N=18, F=512) is 22 times larger, with more than 550M parameters. Despite the differences, our method is competitive and matches or beats DAT in half the datasets.

## 7 DISCUSSION

**Algorithm.** Our results suggest that there are strong potential benefits to using smartly routed pre-trained experts *when* the domain of the experts broadly matches that of the downstream tasks. We have clearly seen this with natural images. Instead, as expected, when there is a skill mismatch (e.g. trying to solve a counting task with diverse single-object recognition experts) we have not observed

any significant gain or loss. Still, in these cases, the expert selector can fall back on the generic model or representation. When there is an extremely relevant expert for a task –say, our *flower* or *plant* models for the Oxford Flowers 102 task–, using full network experts proved beneficial. In contrast, many datasets did not have a perfect match, and adapters seemed easier to fine-tune in these cases.

**Impact.** In the near future, we foresee large computer vision systems composed by a wide range of pre-trained specialist modules. These modules may be based on huge amounts of data, small but high-quality curated repositories, or even on private and proprietary content, and they would cover a diverse spectrum of canonical tasks (object recognition, some way of narrow reasoning, counting, sorting, etc.). Some of them may not even need to be end-to-end learned from data.

**Future Directions.** There are a number of exciting follow-up research directions. Selecting and combining multiple experts for any downstream task is a natural extension of our work. This could be especially useful for tasks that require understanding several concepts, not necessarily captured by a single expert. Per-example routing (i.e. applying routes tailored to each individual data-point) could also lead to improvements based on targeted processing, for example, in the context of tasks with instances of various difficulties. Finally, moving beyond our experts based on label hierarchies, and towards automatic discovering and training of experts could unlock even further gains.

### ACKNOWLEDGMENTS

We would like to thank Alexander Kolesnikov, Lucas Beyer, Xiaohua Zhai and Jessica Yung, who worked on the original BiT, that we use as baseline, and shared their insights with us. Josip Djolonga and Maxim Neumann provided very useful feedback on an earlier version of this work. Finally, we thank whole Google Brain team in Zürich for many useful discussions and support.

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
