# OpenReview forum: "Scalable Transfer Learning with Expert Models"
_ICLR.cc/2021/Conference — ICLR 2021 Poster_

### Official Review · AnonReviewer1 · 2020-10-26
**not enough technical innovation**

**Rating:** 5
**Confidence:** 4

**Review:**

Summary:
The authors propose a four-stage transfer learning strategy:
1. pre-train a baseline model on the entire source data
2. fine tune the baseline model on different parts of the source data (determined by the label hierarchy) to get multiple experts.
3. for the target task of interest, select the best expert based on the nearest-neighbor performance
4. fine-tune the selected expert on the target task
The authors tested their method on two transfer setting: 1) from ImageNet to VTAB 2) from JFT to VTAB.  Empirical results show that their method outperform the baseline.

Strength:
1. The paper is clearly written and easy to understand.
2. Exploiting the label hierarchy to train multiple experts and select the best one makes sense.

Weakness:
1. Maybe I am missing something, but I don't see much new insights from the paper. When doing multi-source transfer, selecting the best performing expert for the target task seems like a straightforward baseline.
2. The method does show empirical improvements, but perhaps not strong enough. From IN21k to VTAB-1k, the full model achieves 71.4 while the baseline achieves 69.8 (2.29% relative improvement). From JFT to VTAB-1k, the full model achieves 70.2 while the baseline achieves 69.8 (0.57% relative improvement).

Other questions:
1. I didn't quite understand what do you mean by "combining all experts" in section 6.5. Is that the same as doing ensemble? If so I think the comparison between "All Experts" and "Baseline" is not very fair.

---

> ### Author Response · Authors · 2020-11-17
> **Reply to AnonReviewer1**
>
> Thank you for your review.
>
> 1. We agree that the goal is (of course) to select "the best performing expert". However, the paper argues that the best way to do this *efficiently* is not obvious. With enough resources/time we may be able to transfer every model to the target task through fine-tuning and then select the best. Unfortunately, this is too expensive in practice in the presence of many upstream models. The key innovation in this paper is to propose a simple yet inexpensive proxy that is able to find the right model (or adapter) from a large pool *without* fully fine-tuning all the models. We compare a number of options, and also to methods such as DAT that use more expensive techniques to build task-dependent experts. We show that the proposed kNN approach performs well across several tasks.
>
> 2. With JFT the best experts model attains +2% absolute, and with ImageNet-21k +1.6% absolute. These are quite large improvements for no (or very little) increase in final model size, since the benchmark is an average across 19 diverse datasets. For context, the authors of the original paper introducing the benchmark [arXiv:1910.04867](https://arxiv.org/pdf/1910.04867.pdf) report only a 2% absolute gain when increasing the network size from ResNet50x1 to ResNet152x2 (10 times more parameters) on Imagenet.
>
> Re: “Other questions 1”: “All experts” means selecting from both the adapter and full network experts simultaneously. We create a single pool with all the models, and use the kNN method to pick the right model. It is *not* an ensemble, and the final network has a computation cost equal (or very similar to) the baseline. This experiment shows that the kNN can successfully select from experts with very different architectures and get the “best of both worlds”. It also shows that kNN is robust to increasing the size of the pool of expert models to a very large number (the largest pool has almost 600 experts). Thank you for pointing out this possible misunderstanding, we will improve the wording in the paper to explain this more carefully.

---

> > ### Comment · AnonReviewer1 · 2020-11-25
> > **reply to rebuttal**
> >
> > Thanks for the clarification. Although I still feel the idea of the paper is not exciting enough, I will increase my score for the extensive & solid experiments.

---

### Official Review · AnonReviewer4 · 2020-10-28
**Expert model provides better representations for transfer learning**

**Rating:** 7
**Confidence:** 4

**Review:**


**Summary:**

This paper presents a novel method for obtaining better representations for transfer learning. Specifically, instead of using a generic representation for various down-stream tasks, this paper proposed to create a family of expert models in pre-training, and selectively choose one expert to generate representation depending on the target transfer learning task. A simple yet effective k-nearest neighbor strategy is used for picking the best-fitting expert. This paper has extensive experiments including pre-training models on two large-scale image datasets, and evaluated on two transfer learning benchmarks (VTAB and datasets from DAT).


**Reasons for score:**

The general idea of this paper (i.e. replacing generic representation with one from target-dependent expert model) is very intuitive, and the experimental validations are very solid. However, the novelty and technical contribution of this paper is only moderate. Overall, I think it's a good paper and may inspire future work on more efficient and effective transfer learning.


**Pros:**

1. The proposed idea is intuitive, and empirically very effective. Different from focusing on architectures for transfer learning, this paper focused on using different representations to improve transfer learning quality. This is complementary with many existing techniques on transfer learning.
2. The experimental validations are extensive and solid, e.g. all reported accuracies have confidence interval so the comparison is more informative.
3. The paper is well-written and easy to follow.

**Cons:**

1. In Section 4.1 two variations of "MoE family" are proposed, i.e. Full ResNet Modules and Adapter Modules. For Adapter Modules, it seems all experts share blocks and only differ in adapter module (Fig.3 b). However, the effectiveness of constructing "expert" model in this way lacks supporting evidence, i.e. how well each "expert" performs on the corresponding/non-corresponding domains?
2. I am wondering if Table 1 should add performance comparison with the baseline model B, so it would be more straightforward whether the expert branch selected by the proposed strategy is more effective on the target datasets.
3. For "All Experts" in Table 2, I find it unclear as in Section 6.5 "Combining All Experts" it doesn't explain how this model works. Does it mean all experts are simultaneously selected, i.e. $x_i := F_i(x_{i-1} + \sum_e a_e^{(i)}(x_{i-1}))$? In that case, if each adapter module introduced 6% parameters (Section 4.1) the extra parameters will be non-negligible.
4. Please clarify: as shown in Figure 3 (a) ResNet-50 has four blocks and adapter module is added before each block; does the proposed system support choosing different adapter module at different block? E.g. for the first two blocks of a specific dataset $a^{(1)}_i$ and $a^{(2)}_j$, is it possible that $i != j$? In other words, the "performance proxy" strategy is applied to each block, or at the end of the entire network?
5. Section 6.1 mentioned that ImageNet21k use 50 experts and JFT use 240 experts, but in supplementary JFT seems to have 244 experts. Also from Table 4 in supplementary C. 6, some dataset choose "baseline" as the selected expert. Does the "baseline" serve as a standalone expert, or a base of all experts (e.g. in adapter modules setting do the adapter serve as residual to the baseline)? There seems to be some inconsistencies here.


**Questions during rebuttal period:**

Please address my questions in the cons section.


**Some typos and minor issues:**

--  Supplementary D.1, "Unconditional pre-training" the last sentence is incomplete.

---

> ### Author Response · Authors · 2020-11-17
> **Response to AnonReviewer4**
>
> Thank you for the thorough review.
>
> With respect to the cons you raise:
>
> 1. The improvement of the adapter experts over the baseline provides some evidence that these models are able to specialize. However, as a more direct measure we can also add a plot where for each slice we show the performance on that slice of its expert model based on adapters and the baseline model. We have done this in the past for full models (not in the paper though), and the difference is quite large. As one would expect, due to the fine-tuning in the slice, the expert models’ performance in their own data slice is much higher.
>
> 2. We will add the non-expert baseline to Table 1.
>
> 3. “All experts” means selecting from both the adapter and full network experts: we consider a pool that it’s the union of both expert pools. It is not an ensemble, and the final network has a computation cost equal (or very similar to) the baseline. This experiment shows that the kNN can successfully select from experts with very different architectures and get the “best of both worlds”. Thanks for pointing out the ambiguity! We’ll fix the wording in the updated version.
>
> 4. Great question. No, we enforce the use of the same expert in every block. Otherwise the kNN would suffer from a combinatorial explosion of paths. However, we are currently researching this interesting direction which requires layer-specific routers which decide which expert to apply for a given input.
>
> 5. Good catch. The reported results were achieved with 240 JFT experts, we will fix the typos in the paper. For the full network experts, the baseline is a standalone model in the pool of experts. In addition, all the other experts are fine-tuned from it. In the case of adapter experts, we also add the original baseline (without adapters) as a standalone model to the pool of experts.

---

> > ### Comment · AnonReviewer4 · 2020-11-23
> > **Reply to author rebuttal**
> >
> > Thanks for the clarification, I'll keep my rating for now and will discuss with other reviewers later.

---

### Official Review · AnonReviewer3 · 2020-10-29
**Good paper**

**Rating:** 7
**Confidence:** 2

**Review:**

The authors address transfer learning scenarios. In particular, the authors resort to training to a diverse set of experts and "cheap" performance proxies to select, for a given task, the relevant expert. This "per-task routing" is conducted via a nearest neighbor classifier based on a reduced representation for each expert. Two variants are considered: (1) full ResNet50 models are used (one for each expert) and (2) "compact adapter modules", which depict expert layers between ResNet block (all experts are learnt simultaneously with the same model backbone).

Comment: The procedure how the kNN proxy works remains a bit unclear in the main text (maybe move some material from the supplemental material to the main text).

Positive:
- On-pair performance with other approaches,  but faster and less parameters
- The authors provide test statistics (i.e., not only single runs)
- The work is well written and well structured)
- Comprehensive supplemental materia

Negative:
- The authors argue that their approach is 500-1000 times faster (main contribution), but this does not become clear in the main paper (also not in the supplemetal material?)
- Minor: Training the full model version (i.e., full ResNet model for each expert) is very expensive from a computational perspective.

Overall, I think this work might be worth being accepted at ICLR. I am not an expert for (recent) transfer learning approaches, so I might be missing something.

---

> ### Author Response · Authors · 2020-11-17
> **Reply to AnonReviewer3**
>
> Thank you for your review and feedback. We will clarify the kNN performance proxy in the main text.
>
> First, as we argue during the paper, the cost of pre-training the pool of experts is amortized over time, since they can be reused when new downstream tasks are learned. Likewise, in Domain Adaptive Transfer (DAT) the cost of pre-training the generalist model on JFT is also amortized. The costs that cannot be amortized are those corresponding to the “expert preparation” step and the downstream fine-tuning. The asymptotic costs of all steps are depicted in Table 8 (Appendix F.3).
>
> Regarding the “expert preparation” step, which is the part that differs between the two methods when learning a new task, it requires less than 2 hours to select an expert model among a pool of 240 experts, based on 1000 images from the downstream task, using a single NVIDIA V100 GPU (see Appendix C.1). This is the real duration of the expert preparation step, following our method. Using the same hardware and image pre-processing, fine-tuning a R50 on JFT for 4 epochs requires ~1800 hours. This is the corresponding cost following the Domain Adaptive Transfer approach.
>
> Since fine-tuning on the downstream dataset takes only a few minutes (see Appendix E.2), it’s impact is negligible (compared to the previous step), and thus our transfer approach learns a new task approximately 900x faster than that of Domain Adaptive Transfer, when pre-training is amortized and same hardware and model specifications are considered. Depending on different optimizations and system bottlenecks, we expect our approach to be 500 - 1000x faster.
>
> Indeed, pre-training the experts is computationally expensive. In particular, training the Baseline (one model on all the data) is most expensive, but also required by any large-scale transfer algorithm, whereas training each individual expert only requires finetuning it on its data slice for a couple of epochs.. In addition, the idea behind this work is that the experts need only be trained *once* (consuming those computational resources), and as we argued before this cost is amortized over time. After this, the user of the experts needs relatively few resources to run the proposed method, pick the best expert, and fine-tune it on their downstream task.
>
> Note: we have open-sourced 48 expert-models trained on ImageNet-21k slices. We will add the link here and to the paper after the end of the review process.

---

### Official Review · AnonReviewer2 · 2020-10-30
**Reasonable Heuristic Method, Thorough Experimental Evaluation**

**Rating:** 6
**Confidence:** 3

**Review:**

Summary: The authors propose to choose the pre-trained model from several experts for better transfer learning performance.

Quality: The method is a heuristic one, but generally simple and reasonable. The authors perform plenty of experiments to show the influence of different design and how the method compares with other methods. Overall, the paper is sound.

Clarity: The paper is well written and easy to follow. I have two questions:
1. Does the number $n$ of adapters equals the number of experts? Or (because there are four blocks) does the number of experts equal the combination of adapters, $n^4$?
2. I have one important question in experiments: Did you compare your method with direct transferring B (the initialization model)? This is the actual baseline to justify the effect of experts and your selection scheme. I think because of residual connection in adapters, B is included in the experts. But still, B could be the best choice but not selected due to the imperfection of selection scheme. I would like to see the results, I may change my score based on that.

Originality: The paper is a novel method for transfer learning, though using multi-expert system/adapters in network are not new.

Significance: This paper could be an inspiring work for transfer learning community. Using multi-expert system for transfer learning seems under explored in deep learning era.

---

> ### Author Response · Authors · 2020-11-17
> **Reply to AnonReviewer2**
>
> Thank you for the positive feedback. We agree that deep experts are under-explored for transfer learning, and an exciting direction.
>
> With regard to your questions:
>
> 1. Great question. In this work we do not explore combinations of adapters from different experts. As each expert is always composed of 4 adapter blocks, we train a total of $4n$ adapters that represent $n$ experts.
>
> 2. Yes, we do include the initial model (trained on the entire upstream dataset) as a baseline in our results. This is the model marked as "Baseline" in Figure 2, Table 2, and Table 3. The baseline is indeed selected by kNN in a few datasets (see Table 4 in Appendix C.6).

---

> > ### Comment · AnonReviewer2 · 2020-11-18
> > **Need Further Explanation**
> >
> > Thank you for your reply. I still need some clarification on the second question for making decision.
> >
> > I was trying to say that, if directly transferring the full model, $B$, yields the best performance, the expert selection step will be non-sense, no matter it is kNN or anything else. If that is true, we do not need any 'expert's, right?
> >
> > So I read the paper again carefully, and I was still not able to find detailed description of the methods in the tables. Does 'Full' refer to fine-tuning $B$?
> >
> > I highly suggest the authors to clarify the setting of each method in the tables. These abbreviations really confuse me a lot.

---

> > > ### Author Response · Authors · 2020-11-19
> > > **Reply to AnonReviewer2**
> > >
> > > We appreciate the quick response to our comment.
> > >
> > > Indeed, if simply transferring the full model *without any specialization* worked best, one would not need expert models. But that is not the case, as shown by the fact that the average performance of this approach (the Baseline) is lower than that of any type of expert model. However, while in a couple of datasets the full model without any specialization may be the best one --when the downstream data resembles well the whole upstream data, for example--, using experts is quite helpful overall, and kNN is able to select the baseline model in such cases (see Table 4 in the appendix).
> > >
> > > “*Full*” and “*Adapters*” refer to the *parameters* of this model that are specialized *to a subset of JFT/ImageNet before any downstream fine-tuning*. These are essentially two alternative architectures for the expert models, both of which originate from the Baseline model. In both cases, we fine-tune (i.e. transfer) the entire expert model selected by kNN on the downstream task.
> > >
> > > Given that after the rebuttal we have an upper limit of 9 pages, we will clarify this in Section 4 and the tables. To doubly clarify the terminology here:
> > >
> > > 1. __Baseline (B):__ This is just transferring the ResNet-50 that was pre-trained on JFT or ImageNet21k. This is the standard practice in transfer learning.
> > > 2. __Full:__
> > >     * The baseline B hat has been *specialized* by further pre-training all parameters of the ResNet50 on a *subset* of upstream data, e.g. pre-training on data about ‘cars’ in ImageNet21k. We have a pool of such expert models, as we generate a separate ‘expert’ by pre-training on different subsets of the upstream data.
> > >     * We compare different methods to select which ResNet-50 expert to use, and show there is a significant improvement when we select such a specialised model for a new downstream task.
> > >         * We note that the baseline model B is part of that pool - if it is beneficial to select it (according to kNN accuracy), our method will choose it.
> > > 3. __Adapters:__
> > >     * The adapters are an architectural adjustment to ResNets that we propose (Diagram in Figure 3, described in Section 4.1). It is a very small component, with approximately 6% of the parameters of the full ResNet-50. Only these extra parameters are specialized to a subset of the pre-training data, the rest are kept as in the Baseline model.
> > >     * Similar to above, we show significant improvements from transferring a single generalist model (baseline). The adapters however allow for particularly efficient training and reduced storage costs.
> > >         * Like with full models, the baseline model B (with no adapters) is part of that pool. Once again, our method can (and does) select it according to its kNN accuracy in some cases.
> > > 4. __All Experts:__
> > >     * We simply include in the pool the two types of expert architectures: “Full” and “Adapters”.
> > >     * This yields the best results, since some architectures are better suited for a particular downstream dataset than others, and kNN does a good job selecting the right one.

---

> > > > ### Comment · AnonReviewer2 · 2020-11-19
> > > > **Reply to Authors**
> > > >
> > > > Thank you for the explanation! I'll hold my positive rating.

---

### Decision · Program_Chairs · 2021-01-07
**Final Decision**

**Decision:**

Accept (Poster)

**Comment:**

The presented idea is aligned with past work using multiple experts or multiple sources for transfer. However, it is positioned uniquely and cleverly in that the approach is developed with scalability in mind. Within this setting, the paper is convincing. Although the approach does not come with strong backing theory, it is intuitive and seems to work well. During the discussions phase, the authors have clarified some questions that made the paper convincing, even if it is a relatively heuristic approach. The results are strong if one is concerned with both quantitative performance and efficiency, a combination of objectives very often encountered in practice. Overall, it is expected that this idea can stimulate further research along those lines, especially since this paper is very nice and easy to read.